A new theory of tree sap flow

Török András 1
Hajduné Kubovje Anikó Anna 2 3
Hardi Balázs 3
http://orcid.org/0000-0002-8733-4286 Bergmann Ralf 3
Szigeti Krisztián 2 3
http://orcid.org/0000-0001-7343-0413 Máthé Domokos 2 3 4 domokos.mathe@hcemm.eu
Hegedüs Imre 2 3 hegedus.imre1@semmelweis.hu
1 HUN-REN Institute of Earth Physics and Space Science , Sopron , Hungary
2 In Vivo Imaging Advanced Core Facility, Hungarian Centre for Excellence in Molecular Medicine , Szeged , Hungary
3 Department of Biophysics and Radiation Biology, Semmelweis University , Budapest , Hungary
4 CROmed Translational Research Centers , Budapest , Hungary
Brygadyrenko Viktor
Electronic publication date: 2025 Jul 31
Publication date: 2025
Volume: 13
Electronic Location ID: e19670
Received 2025 Feb 18; Accepted 2025 Jun 6
Copyright: © 2025 Török et al.
Copyright year: 2025
Copyright holder: Török et al.
License: This is an open access article distributed under the terms of the Creative Commons Attribution License, which permits unrestricted use, distribution, reproduction and adaptation in any medium and for any purpose provided that it is properly attributed. For attribution, the original author(s), title, publication source (PeerJ) and either DOI or URL of the article must be cited.
License URL: https://creativecommons.org/licenses/by/4.0/

Keywords: Water transport in trees, Suction-pressure pipe, Pulsating system, Different time-phase cycle, Aquaporin

Funding: The European Union’s Horizon 2020 Research and Innovation Program 739593 HCEMM, supported by EU Programme H2020-EU.4.a This work was supported by The European Union’s Horizon 2020 Research and Innovation Program, grant agreement No. 739593: HCEMM, supported by EU Programme: H2020-EU.4.a. The funders had no role in study design, data collection and analysis, decision to publish, or preparation of the manuscript.

==============================
Introduction

The theory of water transport in trees, according to which the main driving force of water movement is the suction created by the evaporation of water by the meniscus (the curved surface of the capillary liquid column) on the evaporating elements, supported from below by root pressure, is controversial. The main physics argument against it is that the capillary effect in nature is around 1 m. In the case of open-air gaps, the leaf cannot suck in the water against gravity because, in this case, the plant would not be sucking in water, but air through the open-air gap.

Aim

To present a new theory of three-sap flow and to support it with practical observations, previous data from decades of experimental measurements, and direct measurement data obtained by the authors.

New theory

When evaporation occurs, there is no suction towards the canopy, but pressure is due to a reduction in the cross-section of the water pipe caused by heat loss through evaporation. At night, when evaporation stops, a thermal equilibration process is triggered, restoring the pipe’s original cross-section. This generates suction and draws water from the soil. As the hydrostatic pressure in the pipe is high for tall trees, the pipe is segmented.

Materials and Methods

To study the change in the wood’s cross-section, a mechanical pressure-sensing transmitter-amplifier instrument was used. The instrument is designed to convert changes in the diameter of a millimeter-sized tree into easily detectable data by increasing the diameter by an order of magnitude. We also used high-resolution computer tomography (CT) to measure the cross-sectional image of oak trees to explore areas rich in water.

Results

The experimental results show that the tree diameter increases during the night (suction phase) and decreases during the day (pressure phase). Many measurements in the literature show a similar phenomenon. The CT scan results showed that the outer, living one-ring area of the tree is rich in water, from which passages lead to the passive water storage inside the tree.

Conclusions

Several examples have been given to prove this theory. Water transport is not based on physical mechanical laws alone. Complex physiological, biochemical, and biophysical processes may be behind the operation of the pipe system.

Introduction

From plant anatomy, we know quite well that the dividing tissue of the cambium gives rise to the fascicles, which are dedicated to different tasks, including the cells specialized in water transport, which are transformed into water transport tubes by longitudinal interconnection. Towards the outer sheath of the tree, they form chain cells that transport nutrients. The treatise analyses water transport when water moves in the so-called xylem elements. In vascular plants, one of the two transport tissues is the xylem, or woody part, and the other is the phloem, or limb (Fig. 1).

Figure 1 Cross section of the tree trunk.

The sapwood and heartwood are located inwards from the cambium, and the phloem (its dead part is the inner bark) and outer bark outwards.

The theories of water transport that have been offered so far differ only slightly, with no substantial differences. In these theoretical explanations, the main driving force for the movement of xylem sap is the evaporation of water from the surface of the mesophyll (leaf litter cells) into the atmosphere. Evaporation causes mesophilic cell walls to form millions of tiny menisci (the capillary fluid column curved surface of the capillary). Their surface tension exerts a negative pressure or pulling force on the xylem, which pulls water up from the roots and soil. Root pressure is commonly cited as the downstream driver of water transport. It is supposed that the water potential of the roots is more negative than that of the soil (more precisely, the soil solution). In that case, water can usually enter the roots from the soil by osmosis due to the high concentration of solutes. This creates positive pressure that moves the xylem sap towards the leaves. This phenomenon is known in the literature as root pressure (Bentrup, 2017).

The question arises whether water transport in the tree can work as described. A system will work as intended if it has a functioning mechanism, if there is a matching force of sufficient magnitude acting constantly, and if the structural elements can withstand it.

Let us look at each of these separately.

One of the forces mentioned is the capillary effect, the property of a fluid to move in a narrow, confined space against the force of gravity. If the diameter of the tube is small enough, the combined forces of surface tension and adhesion forces between the tubes will lift the liquid into the capillaries of the plant. However, the capillary rise in nature is around 1 m, and water in capillaries cannot rise higher than this. The root pressure (pushing force) from osmosis can be small or large, it doesn’t matter. it is not suitable for continuous water delivery because it needs to be worked on to keep the water flowing. This requires a mechanism to ensure that the force can move constantly in that way.

The system mechanism is flawed. The only way to deliver water is to squeeze it out of the system, so there is suction and pressure. Water delivery from trees can be compared to the operation of water pumps because similar systems work similarly. There are two types of these pumps regarding how they deliver water.

At the same time, phase suction-pressure pumps, e.g., gear pumps (Fig. 2).

Figure 2 Gear pump.

Suction and pressure in this case occur in the same time phase.

The other is the differential time-phase pump, best known as a suction-pressure well, or the laterally-driven piston pump (Fig. 3).

Figure 3 Piston pump.

In this case, the suction and the pressure occur in different time phases When the piston moves inwards (to the left in the diagram), a pressure drop is created in the space directly in contact with the piston, opening the lower valve and facilitating fluid flow. Then, as the piston moves outwards (right in the diagram), the pressure in the space between the two valves increases, closing the lower valve and simultaneously opening the upper valve, allowing the aspirated liquid to escape upwards.

The operating principles of the two types of pumps are described in more detail below. Water cannot be transported (this is also true for organic systems) by suction or pressure alone. Nor can water be transported by pushing at the bottom and sucking at the top.

In the case of evaporation, the pipes must be saturated with water, because evaporation can only take place in this case. A force balance condition is required for any height. The downward gravitational force must be in balance with the suction force in the opposite direction. For water to enter the leaves at a certain height, a force greater than the suction force creating the force balance is required, depending on the internal resistance. If one wants to illustrate this with a practical example, for example, a beech tree twenty meters high, which can be considered medium height, means that there must be a suction force in the leaves at twenty meters greater than two atmospheres. With two atmospheres of pressure, you can pump the tires of a mid-size car hard. How does the leaf tissue withstand the reverse, a suction of two atmospheres (two bars)? Additionally, that much suction cannot be created. The analysis shows that the theory of water transport as understood so far is flawed. It is faulty because there is not enough force in the system to match the mechanism. So, there is no guarantee that the force will work and move along the path. The mechanism is faulty because water cannot be transported in this way. The biggest contradiction is that at high altitudes, the leaf tissue is unable to withstand the suction generated.

In the context of evaporation, the conducting vessels must be fully saturated with water, as evaporation can only occur under these conditions. A force balance condition must be satisfied at any given height within the plant. Specifically, the downward gravitational force must equilibrate with the opposing suction force. For water to be transported into the leaves at a specified elevation, the force must exceed the suction force that establishes this equilibrium, which is contingent on the internal resistance of the plant system.

To illustrate this with a practical example, consider a beech tree with a height of approximately twenty meters, a representative medium height. This scenario necessitates that the suction force exerted at the leaf level exceeds two atmospheres of pressure. Notably, the pressure of two atmospheres is analogous to the pressure required to inflate the tires of a standard mid-size automobile. This raises the question: how can the leaf tissue endure such significant reverse suction, equating to two atmospheres (two bars)? Furthermore, it is important to note that generating such a level of suction is, in fact, unfeasible.

This analysis indicates that the prevailing theories regarding water transport within trees are fundamentally flawed. The deficiency arises from the inadequacy of forces within the system to facilitate the described mechanism. Consequently, there is no assurance that the requisite forces will effectively propagate along the intended pathways. The mechanism fails because the proposed method of water transport is not viable. The most pronounced contradiction lies in the observation that, at elevated heights, leaf tissue is incapable of withstanding the suction forces generated.

Water potential, the big revealer

However, the concept of tolerability is not a category of proof. The untenability of traditional theories is highlighted by the water potential theory. In 1960, an Australian and an American researcher, Taylor and Slatyer, proposed to use the chemical potential of water as a basis for evaluating the soil-plant-air system, which they argued had a water potential difference (Slatyer & Taylor, 1960) (Fig. 4).

Figure 4 Water potential difference in the soil-plant-air system.

According to the water potential theory, the potential of water decreases and can be as low as −100 MPa in air. The dimension of chemical potential is measured in J/mol and not pressure (MPa). Negative pressure is unintelligible in physics. Image source credit: (C) Carter & Lumen Learning (2025).

It has been proposed that the chemical potential of pure water is defined as zero. The water potential of a system, or of water within a segment of that system, is expressed in terms of pressure and reflects the chemical potential of pure water within the context of the water system. When the water potential of pure water is at its maximum (0), the water potential of solutions is comparatively lower, typically represented as a negative value (Taiz et al., 2015). Water naturally moves from regions of higher water potential to those with lower water potential, similar to how energy in the form of heat disperses from areas of higher to lower temperatures (Sutcliffe, 1979).

In open-ended pipes (stomachs), there is no suction force; therefore, water cannot flow against gravity. If a suction force existed in the leaves, the plant would draw in air instead of water. The concept of water potential addresses the soil-plant-air system and indicates that evaporation leads to suction through the air, effectively pulling water out.

The atmospheric pressure around us, established by Evangelista Torricelli in 1643, limits suction capabilities (West, 2015). von Guericke (2012) demonstrated this with his Magdeburg hemispheres experiment, showing that atmospheric pressure could prevent two hemispheres from being pulled apart despite the removal of air within them. The maximum achievable suction value under terrestrial conditions remains lower than atmospheric pressure, which must be contained. Notably, a low-pressure reading of 860 hectopascals was recorded in a tornado funnel on June 24, 2003, in South Dakota, slightly below the normal atmospheric pressure of 1,013 hectopascals (Lee, Samaras & Young, 2004).

The question arises as to how such high suction values could be detected in tall trees. The measurement was carried out with a Scholander pressure chamber (Fig. 5).

Figure 5 Scholander pressure chamber.

The essence of measuring with a chamber is to place a freshly cut leaf or leaf stem in it with the cut stem facing outwards. The gas pressure inside the chamber increases until liquid appears on the protruding cut end. The gas pressure is then assumed to balance the water potential of the leaf cells. The measurement starts when this sap appears, including the xylem sap and the phloem sap because both tubes have been cut.

This method consists of placing a leaf or a leaf stem in a chamber that can withstand a gas pressure of 5,000 kPa so that the stem or the cut end of the stem protrudes from a locking device fixed to the mouth of the vessel. Then, a gas, such as nitrogen gas, is released into the chamber, and the pressure of the gas is increased until the liquid appears on the cut surface of the plant protruding from the chamber. Then equilibrium between the leaf cells and the sap in the wood, and the gas pressure exactly balances the leaf cells’ water potential (Scholander et al., 1965).

As you can see in the diagram, when the branch is cut, the plant’s sap is retracted. The perceived suction value is read when the sap appears. The most fundamental flaw in the measurement method is that when the branch or leaf is cut off, both suction and pressure tubes are cut through according to the conventional theory. According to the theory, the xylem is a suction tube, but the nutrient-transporting phloem’s rust tube is a pressure tube. This was found by cutting through the mouthparts of sucking insects under the tip bud of the leader drive and pressing out sugary sap. The conventional theory is that a large suction in the leaf should be converted into a small pressure. Such a mechanism to convert a large suction into a small pressure is not found in the leaf. The water transport researchers are very self-critical in pointing out that with this instrument, even though many people have climbed trees with it, they have not been able to detect differences in water potential. The measurements ranged from −0.5 to −0.8 MPa. Zimmerman modified the device, but the suction-pressure tube anomaly persisted (Balling & Zimmermann, 1990)—no wonder the non-existent water potential could not be measured with an instrument based on false principles. The error was caused by the water delivery researchers forgetting to tell Scholander that they had cut both suction and discharge pipes during the stem cutting. If not, how do trees transport water? Let us examine how these contradictions can be resolved and what new mechanism can be found to explain water transport. We will then see if there are experimental results in the literature that support the hypothesised mechanism.

A proposed new water transport theory and its basics available in previously reported experiments

The leaf surface has small openings (stomata). These air slits, which are mostly located on the back of the leaf, provide air exchange and evaporation. During evaporation the air slits are open. At higher energy levels, water molecules are released into the air. Consequently, the water remaining in the tube cools down. A heat dissipation phenomenon occurs. Since it is an organic system, the cells, which are located above each other and are connected by their walls, respond by a reduction in cross-section. Of course, only the elastic annual ring formed in the last year can do this (this is certainly the case for the stalk oak). In this case, we are not talking about suction, but about pressure towards the atmosphere. The tubes can only shrink up to a certain limit. Once the contraction is complete, the breathing openings close. Evaporation and heat extraction cease. A thermal equilibration process is triggered, which slowly restores the original cross-section of the tubes. This increase in cross-section, in turn, generates suction forces towards the roots, which draw water from the soil. Like a medical eyedropper, when a compressed cylindrical flexible rubber with a narrowed cross-section is released, the liquid is sucked into the tube. As a pressure gradient develops in the tube (the hydrostatic pressure gradually decreases from the bottom to the top), a variable suction or pressure force is required, and the tube is sectioned. Thus, the movement of water can be achieved with a force of nearly equal magnitude and a smaller force, depending on the number of cells segmented. This disconnection results in a peristaltic movement. Another important aspect of peristaltic motion is that this form of motion ensures that the system, actuated by a lateral force, can move along a constant path. Therefore, permanent work is also possible.

In the first column of Fig. 6, you can see that the plant is pushing water from segment number two into segment number one above it. (Fp = pushing force, Fs = suction force, Fc = counterforce, which consists of several forces and is therefore not shown). The valve opening condition Fp is greater than Fc. In the pushing process, the counterforce must increase to close the valve. The moment Fp is less than Fc, the valve can close. Pressure can then be applied from segment 3 to segment 2 above, which the previous pressure has reduced. The results are illustrated in the second column (B). The precondition for opening and closing the valve is the same as before. This process is repeated, and the peristaltic motion shown in Fig. 6 is obtained.

Figure 6 The pulsating system in the pressure-suction phase.

In the first column, the plant is pushing water from segment number two into segment number one above it. (Fp = pushing force, Fs = suction force, Fc = counterforce, which consists of several forces and is therefore not shown). The valve opening condition Fp is greater than Fc. In the pushing process, the counterforce must increase to close the valve. The moment Fp is less than Fc, the valve can close. Pressure can then be applied from segment 3 to segment 2 above, which the previous pressure has reduced. The results are illustrated in the second column (B). The precondition for opening- and closing the valve is the same as before. This process is repeated repeatedly, and the peristaltic motion shown here is obtained. The mechanism of fluid flow at night is illustrated by the two columns on the right. Here, the suction starts from the first segment of the third column (A), which sucks water from segment 2 below. The suction continues as long as Fs is greater than, Fc. The moment Fc is greater than Fs the valve closes. The rest of the suction process is like that described for the pressure process. Here too, a peristaltic motion is formed, the result of which is shown in the fourth column (B).

The mechanism of uprooting at night is illustrated by the two columns on the right in Fig. 6. Here, the suction starts from the first segment of the third column (A), which sucks water from segment 2 below. The suction continues if Fs is greater than Fc. The moment Fc is greater than Fs, the valve closes. The rest of the suction process is like that described for the pressure process. Here, too, the peristaltic motion is formed, the result of which is shown in the fourth column (B).

The mechanism for closing and opening valves is illustrated in Fig. 6. A complementary presentation is included to provide a deeper understanding and illustration of the operating mechanisms, using animated slides to show how the pumps work and, based on this, how plants transport water (see Video S1).

In the pressure process, all segments become smaller and smaller during the day, which assumes that there is a shut-off valve at the bottom at the beginning of the pulsation. At night, the valve at the bottom opens at the upstream suction and the whole system becomes thicker and thicker, sucking in water. Figure 7 illustrates the peristaltic movement of the xylem tube during the pressure and suction stages.

Figure 7 A pulsating system.

The peristaltic movement of the xylem tube during the pressure and suction stages.

Experimental plan for investigating the new theory

A peristaltically pulsating suction-pressure system is important, as is the principle of operation of similar water pumps for the same function. Zsuzsanna Rajkovits, a physicist, describes the sinking and rising of a snail octopus, which is J. Verne’s imaginary submarine, the Nautilus. (J. Verne’s imaginary Nautilus was known as the scientific name of the snail octopus (Rajkovits, 2015)). The basic law of bionics, similar systems work in the same way, one assumes the functional similarity between pumps and water pipes. As mentioned, two types of pumps are known (Csizmadia, 2005; Czupy, 2003). One is the same time-phase suction-pressure pump, such as the ordinary well or gear pump (Fig. 2). These pumps are characterized by the continuous flow of water through the inlet and outlet openings, and both ends of the pump are open. In such pumps, the water flow is ensured by a rotating mechanism between the inlet and outlet, which divides the system into two parts. Between the inlet and one-half of the rotating mechanism is the suction, and between the other half and the outlet is the discharge. The other type of pump is where the suction and discharge are not in the same phase. Examples include the well-known suction-pressure well or the less well-known piston pump, operated by a lateral force like a water delivery pipe (Fig. 3). These pumps are characterized by the fact that they draw water into a tank. In this case, the valve between the tank and the suction pipe opens, but the valve between the tank and the air space remains closed. The next step is to squeeze out the water sucked into the tank. The valve opens, the suction is then closed, but the valve between the air space and the tank is opened. From this process, it follows that where the suction takes place, its end is open; where the suction takes place, the suction (tank) end is closed. This is the suction phase with the suctioned water section. For pressure, the end where the pressure is applied is open, and the end from which it is applied (tank) is closed. The conventional theory is that during evaporation, the flow between the inlet (root) and the outlet (breather) is continuous, and both ends of the pipe are open. This is almost reminiscent of the operating principle of a pump with the same time phase. But where is the rotating mechanism that produces suction at the bottom and pressure above? As far as one knows, there is no such thing in the plant. Then the plant still sucks water into a tank by closing and opening valves in a similar way to the differential pumps.

Figure 8 illustrates the movement of water during the pressure and suction stages. In the pressure stage, the piston pump and the water delivery pipe have the upper valve open and the lower valve closed. In both cases, the water volume of the tank is reduced. In the suction stage, the situation is reversed. Here, both the organic water pipe and the piston pump draw water. Therefore, the lower valves are open, but the upper valves are closed. In both systems, the tank water volume increases. As shown in the figure, the water hose and the piston pump operation require a force, a variable volume tank, a valve, and a mechanism. The latter ensures that the force can move in its own way. The presence or absence of evaporation provides force in an organic system. In the case of a piston pump, this is provided by the power source rotating the wheel (manually or mechanically), to which an eccentrically mounted crankshaft is connected. The mechanism is provided by the pulsating motion of the organic water pipe and the piston motion of the piston pump.

Figure 8 The movement of water in the pressure and suction phases.

In the pressure stage, both the piston pump and the water delivery pipe have the upper valve open and the lower valve closed. In both cases, the water volume of the tank is reduced. In the suction stage, the situation is reversed. Here both the organic water pipe and the piston pump draw water. Therefore, the lower valves are open, but the upper valves are closed. In both systems, the tank water volume increases.

Where are the tanks and valves for live wood? The authors will try to answer this question below.

Materials and Methods

Portions of this text were previously published as part of a preprint (https://doi.org/10.20944/preprints202402.1074.v1).

The first author detected the tree trunk diameter change using a homemade device. The instrument is designed and operates as follows: A tight-fitting rubber stopper is inserted into a water-filled sphere-shaped flexible rubber container, and a capillary tube, also tight-fitting, is inserted into a hole drilled in the rubber stopper.

This structure is clamped to the plant using four screw spindles and two plexiglass plates so that the two flexible systems (the wood and the water-filled bicycle rod with the capillary tube) are sandwiched between two rigid materials, the plexiglass plates.

Then, using the wing nuts on the spindles, the first author places the level in the center of the capillary tube, keeping the two plexiglass plates parallel. This allows him to track the bidirectional movement.

The thread pitch of the screw spindle is one millimeter. The displacements in the capillary tube were linear during calibration. At ninety degrees of twist of the wing nuts, the libellee’s elevation changed four centimeters, at one hundred and eighty degrees of twist, eight centimeters, at two hundred and seventy degrees of twist, twelve centimeters, at a full three hundred and sixty degrees of twist, sixteen centimeters, so that at a one-millimeter change in plant size, the vertical elevation of the horizontal elevator was sixteen centimeters.

The instrument has converted the horizontal movement of the tree into an actual dipping change in vertical movement. (In the case of thick bark trees, dipping changes in the tree presumably do not follow dipping changes in the water pipe because the movements are absorbed by the elastic band).

The measurements were made at a constant temperature on a ficus tree three centimeters in diameter. The constant temperature ensured that the length of the screw spindle connecting the plexiglass sheets could not change due to the constant temperature and thus could not result in measurement error (see Video S2) (Török, 2019).

The CT image was acquired with Siemens Naeotom Alpha Computer Tomograph (90 kVp, 75 mAs, 0.6 pitch, revolution time 0.25, slice thickness 0.2 mm) (Siemens Medical Solutions USA, Inc., Malvern, PA, USA).

The water potential difference between the dry outer part and the wet inner part of the tree trunk was measured by a Digital Multimeter DT-9205A (BERI MáTé EV, Koronco, Hungary).

Results

Localization of the tanks and valves of the water transport system in previous experimental results

The question is raised for the organic system because it is obvious for the piston pump. One tank is the volume difference due to the expansion and contraction of the water transport system. This can be demonstrated with an instrument developed by the first author, and the movement of water can be visualized. The instrument’s construction and operation principles are detailed in the Materials and Methods section. This allowed the authors to follow the bidirectional movement. The pitch of the screw spindle is one millimeter. The displacements in the capillary tube were linear during calibration. At 90 degrees of rotation of the wing nuts, the dragonfly pitch changed by four centimeters, 180 degrees eight centimeters, 270 degrees twelve centimeters, and a full 360 degrees of rotation by 16 centimeters. Thus, the vertical increase in the water level was 16 cm for a one mm change in plant size (see Video S2). The instrument converted the horizontal movement of the tree from an actual change in diameter to a vertical movement. In the case of thick-barked trees, the changes in tree diameter presumably do not follow the changes in the diameter of the water pipe because their movement is absorbed by the elastic strap. Further experimental results measured by this instrument are available in the article of Kökény (2022).

The measurements showed that the tree diameter increased during the night suction phase, and the water level increased in the capillary tube. During daytime evaporation, the displacement was in the opposite direction. It should be noted that the length of the day in plants is from the opening of the air gap to its closure. The plan closes its respiratory aperture in the early afternoon on hot summer days. It switches to night mode. This was also observed in the measurements, because there was a reverse displacement of the water suction nozzle at noon, so the diameter of the tree increased because it was sucking water. No sign of condensation was detected if a nylon bag was put on the branch at this time. The early afternoon air gap closure is well known among plant life scientists. It will be discussed later that suction is a slower, energy-saving process. Hence, frequent daytime aeration is an important factor in plant water management.

Many similar measurements can be found at https://treewatch.net, where you can see online live sinusoidal curves of tree diameter changes and sap flow from different measurement sites in several universities and research institutes across three continents (Europe, Africa, and Asia) (2014–2025).

Localization of the water reservoir in trees using imaging

However, there is another reservoir in the tree, the existence and location of which have been detected by a research team organized by Csilla Béres in the project at village Síkfőkút, using a portable computer tomography (CT) and a high-resolution magnetic resonance imaging (MRI) scanner (Béres et al., 1998). The authors of that report also made a high-resolution CT scan of a cross-sectional image of the sessile oak. The current authors have also repeated this CT scan of a cross-sectional image of oak using a modern, high-resolution CT scanner (Fig. 9).

Figure 9 Cross-sectional view of sessile oak.

The image was acquired with Siemens Naeotom Alpha ComputerTomograph (90 kVp, 75 mAs, 0.6 pitch, revolution time 0.25, slice thickness 0.2 mm). In the picture, the light region contains a higher water concentration, and the dark areas are drier. The active water and nutrient transport are concentrated in the outermost, thin, lighter one-ring area. The lighter areas in the interior are richer in water, but there is already dead wood here, so there is no active transport. The image was taken in May when the trees still contain water left over from winter water storage. The dark part has already run out of water from the winter water storage, it is narrower here and thicker in summer. (This is why the sessile oak felled in winter sinks in the water, in summer it floats on top of the water). The bark is not visible because it does not contain water either, so it is not separated from the dark background. The parenchyma of the bark is also visible in the form of bright lines running towards the center of the tree. Active movement of the xylem tube also promotes nutrient transport in the opposite direction.

Figure 9 shows the obtained CT image cross-section plane of the trunk of a sessile oak. The figure is informative because it tells us the pulsation system is forming water reservoirs inside the trunk using channels. The image shows a narrow circular ring to the canals and water reservoirs. These results replicate and confirm Csilla Béres’ early findings (Béres et al., 1998). The tree was tested within 6 h of felling. In addition, an average voltage difference of 72.8 mV was measured between the dry, outer and wet, inner parts of the tree, even 6 h after felling.

Demonstration of the nutrient transport mechanism using a fig tree leaf

There is currently no single explanation for the mechanism of nutrient transport in plants (see Discussion: The mechanism of nutrient transport). According to the new theory, the pulsating motion of the xylem tube pushes the glucose out of the sieve tube through the tiny pores. The nutrient-transporting sieve tube has an inverse motion to the xylem tube. This is illustrated by the following phenomenon: a leaf was broken off from a fig tree (Ficus carica) at the base of the petioles and then turned vertically upwards towards the sky. Pressing the broken part, latex sap drips out, like herbaceous plants such as bleeding dovetails or dog milkweed. Then, when the leaf was cut down towards the leaf plate, the same thing happened. That is, the sap squeezed out by the xylem tube remained pressed inside the space between the segments formed by the two transport beams (see also Fig. 7). The theory is supported by this experiment, carried out after dark around the time of the longest day. The flow of sap continued until midnight, after which it stopped. That is how long it took for the sap to incorporate. (For further explanation, see Discussion: The mechanism of nutrient transport).

Discussion

The early experiment using CT by Béres et al. (1998) also shows that the plant can use water from the water reservoir, the stored water, and return it to the active pulsating phase. It follows that there is water movement inside the tree even in the already-dead part (Béres et al., 1998). The movement is presumably controlled by the pulsation system. The experiment of Béres et al. (1998) also involved placing isotopes in the tree and using scintillation detectors at different heights to follow the isotope path. They found that the flow was not uniform. In addition to the fast flow, there was a much slower flow, and this persisted throughout the night. At night, the air vents are closed. Csilla Béres also noted that the pull of evaporation cannot explain this phenomenon. It is also interesting to note that the rapid flow in the tracheae formed in the very narrow outer band of the tree in the last year of the tree. They called this free water, and the water in the canals and water reservoirs bound water, either to a material or a structural element (Béres et al., 1998). Given that the deposition and extraction do not occur at the same height, we can conclude that water flow against gravity can also occur in the already dead wood. The mechanism of entry and removal from storage is completely unknown in physiology. At the end of the last millennium, water transport experiments at Sikfőkút were a milestone in physiological research, involving several famous institutions and universities (Tóth, 2013). Without being exhaustive, ATOMKI Budapest, Hungary, DOTE Debrecen, Hungary, University of Marburg, IATA Florence, University of Forestry and Woodworking Sopron, Hungary. The research at Síkfőkút has been reported in several articles by the institutions listed (Béres et al., 1989, 1998; Raschi et al., 1995; Tognetti et al., 1996).

Another important question is where the valves are. The valves are located on the surface connecting the ridge walls of young 1-year-old cells. Their existence has long been known (called transmembrane protein), but their function was unknown. Aquaporin, the membrane water channel, was discovered by Peter Agre and Roderick MacKinnon. (They were awarded the Nobel Prize in 2003). This transmembrane protein molecule allows valve closure and valve opening (Agre, 2004). It also has the last link in the chain, the valve, without which the pulsating system would not work (Fig. 10).

Figure 10 Schematic diagram of a biologically active membrane and the location of the proteins in it.

Transmembrane proteins like aquaporin can transport water from one side to the other of the membrane. In trees, starting or stopping the flow of water through a valve can be as simple as using aquaporins.

To give a live demonstration of this water delivery option, a model has been created, consisting of flexible pipes and check valves with an internal diameter of 2 mm. The manometer in the model shows increased pressure. In the attached video, the first author uses this model to demonstrate the mechanism of water delivery to trees, night and day (see Video S3).

The mechanism of nutrient transport

The physiological literature explains the mechanism of nutrient transport in many ways. The best-known theory is that there is a potential difference between the place of loading and the place of use, based on the phenomenon of osmosis. This results in a turgor pressure gradient, which drives the sap by mass flow. However, calculations have not always confirmed the existence of a gradient. The probable reason is the high internal resistance. Other mechanisms were thought to be involved in the formation of the flow. It is theorized that the pulsating motion of the xylem tube pushes the glucose out of the roasting tube through the tiny pores. The nutrient-transporting roasting tube has an inverse motion to the xylem tube. This is supported by simple experimental results (see Results, experiment with fig tree) that can be replicated by anyone (see also Fig. 7).

Evidence for the water transport theory

(1) The plane-root isotope experiments have clearly shown that the conventional theory is wrong. In his summary article, Tóth (2013) mentions that the transport of water by trees has been shed in a completely new light. His findings were decisive, as they have already been described. Upward flow was also observed at night, which is impossible for closed breathing holes. However, some writers explain the night flow by root pressure. This is incorrect because a poplar’s stake shoots out even if it has not yet formed roots. Roots are formed only after the shoot has been established. The physiological reasons for this will be explained later.

(2) Figure 11 shows time-dependent velocity patterns in the trunks of different tree species, a day after watering. On the vertical axis, K = 0.206 U0.814, where U is the flow velocity/m/min (Németh & Béres, 2016; Raschi et al., 1995; Tognetti et al., 1996). In the figure, the small sawtooth shape may indicate when water in the still organic xylem tube of the annual ring formed in the last year flows through valves from one cell to the cell above. The pattern in the image reflects the water flow from cell to cell. As it is a flow in an elastic system, there is no uniform motion. The opening and closing of the cell valves involve the storage and recovery of kinetic energy. In the early afternoon, the flow decreases. This could mean, as described earlier, that the plant has started to take up water and store it in the woody debris. This is when one noticed a change in the direction of the water level of the first author’s diameter recording instrument, i.e., the tubes were swollen. If you put a nylon bag on the branch at this time, you will not experience condensation. That the plant could have done this could be explained by the fact that the xylem tubes had already used up a lot of water and could move it in another direction. This phenomenon is particularly important in thin-corted tree species such as beech. If there is enough water in the soil, the tree does not shed its bark in the open, because it switches to night mode, even several times a day. Moisture circulation does not stop, and cooling is achieved by suction from the roots. If there is not enough water in the soil, it will immediately drop its bark in the open. This is the phenomenon of peeling.

In addition, the same sinusoidal time pressure curves and the tiny sawtooth-like pattern can be studied on the treewatch.net portal maintained by the Ghent University Laboratory of Plant Ecology in Belgium, where measurement sites around the world are also reported. Thus, for further support of the theory of the temporal evolution of tree sap flow, readers are invited to visit http://treewatch.net (TreeWatch.net, 2025).

(3) The law of continuity proves the theory correct, although Fig. 11 shows the image of a function of a half-sine curve. One thought is that the nocturnal part of it could be found somewhere. Csilla Béres showed it on her computer before her death in Szombathely in the early 2010s (Németh & Béres, 2016). The other half of the sine curve was also plotted. This was important because it gave us the right to invoke the law of continuity. The average speed is higher during the day and lower at night. According to the law of continuity, a high speed has a small cross-section, and a low speed has a large cross-section. (The amount of inward and outward movement can distort this). So, the tree’s casing is moving. Further mobile computer tomography (MCT) measurements by Németh & Béres (2016) mentioned above confirm beyond any doubt that the rate of tree sap flow shows a regular periodic diurnal oscillation and that a 24-h periodic sinusoidal curve can be perfectly fitted to this rate variation (Tognetti et al., 1996) (Fig. 12).

(4) One of the most mysterious secrets of tree life: how does life start in spring, how does a dormant shoot in a bud turn into a leafy shoot? Let us not forget that, at this stage, there is neither evaporation nor photosynthesis. It seems, in autumn, the leaves fall off the tree so that the pressure remains in the flexible transport system. This is easy to see, as water constantly flows upwards at suction and pressure. At the end of the winter, some of the living elements are swollen in the tubes, and thus the water is released from the pressure tube and combines with the sugary compounds next to it. This pressure and excess compound are a prerequisite for life to restart. The diameter of the fold is then reduced. This phenomenon, which is suspected, was pointed out by Kökény (2022). His article can be read in the March 2022 issue of the Forestry Journal. The article is titled Tree Moisture Cycle, Twitter, and YouTube (Kökény, 2022).

(5) The term root pressure is often mentioned in literature as an important factor in water transport. In the first Author’s opinion, there is no root pressure. As described earlier, a one-and-a-half-meter-long stake can sprout without roots, but only if the last living annual ring is full of life. A prerequisite for the onset of life is the partial death of the previous living growth ring, which reduces the plant’s cross-section. Of course, all xylem elements cannot die simultaneously, because then the plants’ growth is not guaranteed. The time interval for the death of the remaining xylem tubes is the first 2 to 3 weeks of May. This is when the bark separates from the woody body, and water appears under the bark. This is when the loggers cut the bark. Thus, from the time of defoliation until the first 2 weeks of May, there are two vintages of live xylem elements in the tree. The floodplain foresters store the poplar and the willow cuttings in a cool, humid place and plant them immediately after delivery. It is true, however, that the delivery system has a variable pressure rhythm. This pressure is highest in the morning and lowest in the evening. There is also an annual rhythm. It is highest in the winter, during the dormant season, when the tree is saturated with water. The Turkey oak that is thrown into the water then sinks. The inner water reservoirs of the tree are almost saturated. It was experimented on in the birch tree before the first author’s house. Before the buds had set in winter, a ten-centimeter-thick branch was cut off and smeared with a liquid, gel-like wound resin. Water dripped down the cross-section of the bush when it budded, but there were also areas where the wound resin bulged out. An air bubble formed and later burst. This phenomenon also proves that wood has a large water reserve in winter and that the wood body is under pressure. This daily and yearly pressure rhythm of the transport system applies to the root and the whole transport system. The rootless sprouting of the stake cuttings and the spring emergence of life can be traced back to the same analogous phenomenon. In some tubes, the living elements are swollen, causing the water to escape from the pressurized tube and merge with the sugar solution next to it. The diameter of the stem is then reduced since the excess material is realized in the leaf shoot. The excess compound flows out of the wood when the branch is cut off. This phenomenon is observed in walnuts and birch. The phenomenon of guttation is also due to the pressure of the transport system. This is when plant fluid is forced out through the hydathodes on the leaf plate. It is no coincidence that this phenomenon occurs late at night in closed breathing openings when the pressure of the transport system is high. It is particularly common in the tropics, where high humidity is common in the crow flies. So, it’s not root pressure either.

In Fig. 13, only the dead water transport cells of various shapes are visible. The negatives of the dead protein molecules across the membranes are visible. There is no need for organic transport tissues inside the tree, but solidification is needed. It is particularly important to achieve high bending strength.

(6) A team of Hungarian, Finnish, and Austrian researchers has used an infrared laser scanner to investigate what happens to tree canopies when night falls. The scientists concluded that the trees were asleep. They used a laser scanner to create a highly accurate and detailed model of the trees they studied, which they repeated every hour at night. It was confirmed that the branches and leaves of the trees were bent down by up to ten centimeters at night. According to András Zlinszky, a biologist at the Hungarian Academy of Sciences, they reached their lowest position just before sunrise and then returned to their previous position within a few hours at dawn (Zlinszky & Barfod, 2018; Zlinszky, Molnár & Barfod, 2017). It is not yet known whether the trees were woken up by the sun or by their internal rhythm independent of the sun (Eetu Puttonen) (Puttonen et al., 2016). The research team measured the shift in old birch trees. The scientific name of the birch is Betula pendula. The name is derived from the Latin word pendere, which means drooping. The branches of the old birch tree hang long. It seemed that nothing proves the theory more clearly than the sleep movement of trees. At night, as water seeps up from the root, the hanging branches become heavier and heavier from the water flowing into the transport system and water tanks, so they bend steadily downwards (Zlinszky, Molnár & Barfod, 2017). During the day, the process is reversed when the pressure is released. The rapid rise is the difference between daytime and nighttime force play. Later, Zlinszky & Barfod (2018) repeated the experiment with exotic tree species a few meters in height and a few centimeters in diameter. The measurements were carried out in a semi-enclosed greenhouse in Balatonfüred. They found that, unlike birch trees, these trees move their foliage up and down (Zlinszky & Barfod, 2018). This movement may be related to the water saturation of the trees, as they correctly guessed. Csilla Béres’ MRI experiment shows that the tree stores water in different places through channels during the night (Béres et al., 1998, 1989). This causes a continuous change in the bending moment in the cross-section of the tree, causing the branches to move up and down. This has been called the pulse of the tree. That is why it was deducted that not all trees are asleep, but they all have a pulse.

The first author has met András Zlinszky several times, so he knew this theory and referred to it as the author who described the water flow in wood with different time phases of suction and pressure (Török, 2022; Zlinszky & Barfod, 2018).

(7) Why do the trees make music? Reading books on plant biology and notes, one can conclude that there is no plausible explanation for the air exchange in trees. An interesting engraving of a succulent plant leaf can be found in the plantarium of János Bognár (Bognár, 2012).

In this section, the valves were distinctly outlined on the stomatal crypt vault, connected to the spongy parenchyma cells, the double vessel ray, and the columnar parenchyma (Fig. 14). The double vessel ray pulsates. As the leaf vessels become thinner, the oscillation amplitude decreases, but the frequency increases. The vibration is transmitted through the spongy parenchyma to the vault of the crypt, where it vibrates. This causes air to flow in and out of the valves. This vibration, caused by the movement of the valves, is converted into sounds that the human ear can hear by an instrument called “music of the plants”. This music was played at two bus stops in Buda. You can find it online by typing the following into your computer’s search engine: “The music of the plants at two bus stops in Buda” (Fürdős, 2021). One feels that solving the mystery of the trees’ making music is one of the proofs of the theory.

(8) Fictional evidence. How does a Scots pine absorb water from frozen ground? Forests usually plant 2-year-old trained Scots pine seedlings with roots and stems about twenty centimeters long. At this depth, this layer of soil freezes in colder winters. The roots then have to take up water from the frozen soil, as the pine evaporates even in winter. The skater slides on the water because the pressure and friction caused by the movement have melted the ice. The authors believe pulsating motion is true for the entire transportation system. It may also be true that as you get closer to the thinner roots, the amplitude decreases, but the frequency increases, which may be greatest in the capillaries. The resulting vibration may then thaw frozen soil particles and absorb water from the plant.

The theory of water transport in trees, according to which the main driving force of water movement is the suction created by the evaporation of water by the meniscus (the curved surface of the capillary liquid column) on the evaporating elements, supported from below by root pressure, is flawed. The capillary effect in nature is around 1 m. In the case of open-air gaps, the leaf cannot suck in the water against gravity because, in this case, the plant would not be sucking in water, but air through the open-air gap. If not, how do trees transport water (Török, 2019)?

When evaporation occurs, there is no suction towards the canopy, but pressure is due to a reduction in the cross-section of the water pipe caused by heat loss through evaporation. At night, when evaporation stops, a thermal equilibration process is triggered, restoring the pipe’s original cross-section. This generates suction and draws water from the soil. As the hydrostatic pressure in the pipe is high for tall trees, the pipe is segmented. To prove our theory, the authors have given several examples. Water transport is not based on physical or mechanical laws alone. There may be complex physiological, biochemical, and biophysical processes behind the operation of the pipe system (Török, 2022).

Figure 11 Water flow was measured in Turkey oak, sessile oak, and hornbeam trunks the day after irrigation.

(SO, sessile oak; HB, hornbeam; TO, Turkey oak). On the vertical axis, K = 0.206 U0.814, where U is the flow velocity/m/min (Németh & Béres, 2016). In all three cases, the change of the flow as a function of time takes the shape of a half-sine curve. (A depression is visible near the apex of the semi-circular curve. This suggests that the plant has closed its stomata for a shorter period at noon, entering a nocturnal mode. This is when it started to suck up water).

Figure 12 Daily change of sap flux of Q. cerris and Q. pertraea measured on 22–25 June 1996 by Tognetti et al. (1996).

These experimental results illustrate that the water flux in the tree behaves according to a one-day periodic time course of the sinusoid curve (Tognetti et al., 1996). At night, the tree’s water flow rate decreases and increases during the day. According to the law of continuity, this is only possible if the diameter of the transport tubes changes. This implies that the diameter of the tree’s sheathing needs to change to a barely perceptible degree. At night, the stoma openings are closed, so there is no evaporation. At night, the xylem tubes dilate, increasing their internal diameter and reducing the speed of water flow. During the day, the stomatal openings are open, so there is evaporation and the water flow rate in the trunks of the trees increases. Mechanically, this means that the xylem tubes become constricted, allowing a faster flow. Here, too, a depression is observed at the apex of the sinus curve, suggesting that the stoma opening closes during the day and the transport system switches to night mode and sucks. (For the plant, daytime is when the stoma is open).

Figure 13 Dead xylem cells from various tree species.

The negatives of xylem elements that have been alive for a year are visible, which are different on the parietal wall (pel).

Figure 14 Stomacript.

The upper columnar part of the section is the columnar parenchyma. It is also connected to the double vessel limb (xylem and phloem), the spongy parenchyma, and the cryptic vault of the stoma, which shows valves. The lower the amplitude, the higher the frequency. When it vibrates, the air moves in and out and makes a sound. The music of the plant device has converted this vibration into sounds that the human ear can hear.

Epilogue

In conclusion, bionics, an interdisciplinary field, seeks to translate biological solutions into technical applications, drawing on the idea that natural selection offers optimal solutions. It posits that similar systems function similarly due to comparable mechanisms. In the context of a tree’s water transport system, mechanisms such as suction-pressure and warming-cooling play critical roles. If any opposing effects are absent, the system may fail.

Unlike water pumps, which lack this complexity, plants adapt to changing climates and water availability. Thus, organic systems exhibit more intricate mechanisms, where physiological and biochemical processes influence water movement. The last living cells in a tree store and release water from nonliving vessels, further complicating water transport.

Ultimately, trees transport water through peristaltic pulsations of suction and pressure, similar to how humans do. This parallel may not be coincidental.

Conclusions

The theory of water transport in trees is controversial. When evaporation occurs, there is no suction towards the canopy, but pressure is due to a reduction in the cross-section of the water pipe caused by heat loss through evaporation. At night, when evaporation stops, a thermal equilibration process is triggered, which restores the original cross-section of the pipe. This generates suction and draws water from the soil. Since the hydrostatic pressure in the pipe is high for tall trees, the pipe must be segmented.

The new theory already has some practical implications, which could even lead to economic returns. (1) End-of-summer watering can be very important when planting roadside trees and fruit trees. Without rainfall, the water reservoirs do not fill up, and the tree will not sprout the following year and will wither.

(2) In trees with full water tanks felled in winter, microbial degradation processes do not start in sawmills during storage. On the other hand, trees harvested later in the spring have to be pulled under a layer of water in sawmills to avoid this degradation, which involves significant additional costs. The explanation for this phenomenon was previously unknown.

(3) Ecotypes are highly adaptable in terms of water consumption. This is discussed in the article.

Almost all of the claims that form the basis of the theory have been confirmed in our study: (A) It can be recognised as a fact because it has been experimentally shown that (1) The theory underlying the daily periodic sinusoidal change in diameter exists;

(2) And the water flow rate also has a daily rhythmic sinusoidal variation;

(3) There is also evidence of an active connection between the outer living tree ring and the water storage inside the tree trunk (see CT image).

(B) Assumptions that can be logically proven that (1) Water transport in the tree must be segmented to work;

(2) One-way valves must be present for the water supply to work;

(3) It is only indirectly possible to prove that the stomatal apertures of the leaves also vibrate during their operation.

(C) Other assumptions that require experimental verification are (1) The aquaporins perform the function of one-way valves in the water transport system;

(2) The roots also vibrate when water is taken up.

(D) We do not know (1) the control of the water delivered by the external active mobile xylem to the internal wood body.

(2) Its function is also unknown.

Supplemental Information

Supplemental Information 1 The pulsating system in the pressure-suction phase.

Animated illustration of the suction-pressure system when the stoma is open and closed.

Supplemental Information 2 Potential difference - oak cross section.

Measurement data of the potential difference in the cross-sectional plane of stemless oak between the outer dry and inner wet areas

Supplemental Information 3 Device for accurate tree branch diameter measurement.

This animation shows the diameter measurement of a tree branch in action, making it easier to interpret the data.

Supplemental Information 4 A working model based on the theory of water transport in trees.

András Török shows how the model works and how water transport can work for trees based on the model.

The authors would like to thank János Gerdenics for their kind help in creating the model presented in the Supplemental Material video, which illustrates the theory in action. Thanks also to Sándor Siffer, who provided us with a freshly felled oak tree from his forest.

Additional Information and Declarations

Competing Interests

The authors declare that they have no competing interests.

Author Contributions

András Török conceived and designed the experiments, performed the experiments, analyzed the data, prepared figures and/or tables, authored or reviewed drafts of the article, and approved the final draft.

Anikó Anna Hajduné Kubovje performed the experiments, analyzed the data, prepared figures and/or tables, and approved the final draft.

Balázs Hardi analyzed the data, prepared figures and/or tables, authored or reviewed drafts of the article, and approved the final draft.

Ralf Bergmann analyzed the data, prepared figures and/or tables, authored or reviewed drafts of the article, and approved the final draft.

Krisztián Szigeti analyzed the data, authored or reviewed drafts of the article, and approved the final draft.

Domokos Máthé conceived and designed the experiments, performed the experiments, analyzed the data, prepared figures and/or tables, authored or reviewed drafts of the article, and approved the final draft.

Imre Hegedüs conceived and designed the experiments, performed the experiments, analyzed the data, prepared figures and/or tables, authored or reviewed drafts of the article, and approved the final draft.

Data Availability

The following information was supplied regarding data availability:

The 3D animations and a film of a working model are available in the Supplemental Files.

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
