# Peer review of "A new theory of tree sap flow"

_PeerJ, doi:10.7717/peerj.19670_

## Round 0.1 · original submission · Major Revisions

· Academic Editor

Major Revisions

Dear authors, the article has aroused great interest among the reviewers and even more questions. I ask you to carefully correct the shortcomings of the manuscript and respond to the fundamental comments. I hope that the new version of this article will be approved by the reviewers.

**Language Note:** The review process has identified that the English language must be improved. PeerJ can provide language editing services - please contact us at [email protected] for pricing (be sure to provide your manuscript number and title). Alternatively, you should make your own arrangements to improve the language quality and provide details in your response letter. – PeerJ Staff

Reviewer 1 ·

Basic reporting

The manuscript is clearly written, using professional and generally unambiguous English, although minor improvements in sentence structure could enhance readability. The introduction and background sections provide sufficient context and thoroughly reference relevant literature, highlighting clearly how the proposed theory fits into existing debates on water transport mechanisms in trees. The manuscript adheres well to professional article structure standards, with a logical flow from introduction through methods, results, discussion, and conclusions. Figures are relevant, adequately labeled, and effectively complement the textual explanations, although some could benefit from more detailed captions or clearer annotations. The raw data and supplementary material are provided, ensuring transparency and reproducibility.

Experimental design

The article presents original primary research clearly within the scope of the journal. The research question concerning the validity of traditional sap flow theories is relevant, clearly articulated, and addresses an important gap in the understanding of physiological mechanisms behind water transport in trees. The experiments conducted demonstrate rigor and high technical standards, involving innovative mechanical sensors and high-resolution imaging techniques such as CT scans. The methodologies are described in adequate detail, allowing for replication, though some descriptions, particularly of experimental setups, would benefit from additional clarification.

Validity of the findings

The validity of the findings is supported by robust experimental data, clearly provided in the supplementary materials and raw data files. The statistical presentation, however, could be more explicit to clearly demonstrate the robustness of findings. The presented results support the authors’ alternative theory of water transport, and the conclusions drawn directly address the original research question without exceeding the bounds of the experimental evidence. It is recommended, however, that the authors provide clearer delineation between established facts and proposed hypotheses to avoid potential overstatements.

Additional comments

Overall, this manuscript offers an intriguing and potentially valuable new perspective on tree sap flow mechanisms. The authors successfully combine practical observations with theoretical insights and historical experiments. The strengths of the manuscript lie in its thorough exploration of previous theories' limitations, innovative use of imaging techniques, and clearly outlined experimental approach. Nevertheless, several areas could be enhanced:

Figures: While figures are informative, their labeling and descriptions can be improved to facilitate better understanding without extensive reference to the text.

Data Interpretation: More explicit quantitative analysis could strengthen the clarity and persuasiveness of the results.

Practical Implications: The manuscript could benefit from a brief discussion of practical implications of the new theory for broader ecological or forestry practices.

Validity of the Findings
The manuscript makes significant contributions toward understanding plant physiological processes, providing credible experimental backing for its proposed theory. While the novelty of the findings is not a primary review criterion, the manuscript offers meaningful replication and validation of past research results, clearly identifying benefits for ongoing research into tree physiology.

In conclusion, the manuscript is well-prepared and makes a solid contribution, subject to minor clarifications and improved figure presentation.

Cite this review as

Reviewer 2 ·

Basic reporting

It seems to be an interesting idea that peristaltic motion due to shrink or swollen of stems or their components resulting from open and closure of stomata could be the main driving force of sap ascent during the day when stomata are open or water suction from soil when stomata are closed. However, I do not fully understand the manuscript including text, figures and videos. Also, I think more evidences are needed on that the peristaltic motion drives sap flow in tall trees. The discussion in this manuscript on the role of root pressure in raising sap is incomplete. Some plants do have rather high root pressure such as in tall tropical bamboos, which is able to drive ascent of sap in night to their canopy. Nevertheless, I think this manuscript might stimulate discussion and further research on the very important topic of sap transport in woody plants and its theory after an adequate revision and probably additional research on tall trees. The language expression of the manuscript needs a substantial improvement.

Experimental design

Additional reseach on peristaltic motion driving sap flow in tall trees might be needed.

Validity of the findings

Need more evidence on that the peristaltic motion due to shrink or swollen of stems drives sap flow in tall trees.

Additional comments

Lines 318-319, “It should be noted that the length of the day in plants is from the opening of the air gap to the air partition.” What means air partition here?

Lines 454-455, “some of the living elements are absorbed in the tubes” ,and Line 483 “In some tubes, the living elements are absorbed” what means living elements here?

In the caption of Figure 6, “The mechanism of uprooting at night is illustrated by the two columns on the right in Figure 6”, what means uprooting here?

Video 3 should better be narrated in English instead of a native language.

Cite this review as

Reviewer 3 ·

Basic reporting

The article text is not everywhere clear and technically correct

The article included sufficient introduction and background

The structure confirmed to an acceptable format

Figures are fine

Experimental design

Original primary research within Aims and Scope of the journal.

The submission did not clearly define the research questions

Rigorous investigation performed to a technical and ethical standard

Methods had not to be described with sufficient detail and information

Validity of the findings

Impact and novelty are not assessed sufficiently

Data analysis is sufficiently presented

Conclusion should be presented in more detailed form

Additional comments

New method is not well revealed

Cite this review as

---

## Round 0.2 · accepted · Accept

· Academic Editor

Accept

Dear Dr. Hajduné Kubovje and Dr. Hegedüs, I congratulate you on the acceptance of this article for publication. I think this article will generate a lot of discussion among readers and interest in the media.

Reviewer 1 ·

Basic reporting

All recommendations have been taken into account by the authors. I recommend the article for publication.

Experimental design

All recommendations have been taken into account by the authors. I recommend the article for publication.

Validity of the findings

All recommendations have been taken into account by the authors. I recommend the article for publication.

Additional comments

All recommendations have been taken into account by the authors. I recommend the article for publication.

Cite this review as

Reviewer 2 ·

Basic reporting

The authors have responded my previous review comments in the revision.

Experimental design

The experimental design seems logical but requires further tests in large trees.

Validity of the findings

This is bascially a conceptional paper that needs further tests.

Additional comments

Line 24, three-sap flow should be changed to tree sap flow.
I suggest the authors to further throughly check whether there are more spelling errors and language problems in the manuscript.

Lines 450-451, “some of living elements are swollen in tubes” may be changed to “some of hydraulically functional tubes are swollen”, as the functional tubes even in the first-year tree-ring are actually all dead cells although there are live parenchyma cells. Therefore, when describing hydraulic function of tubes in annual tree rings in the text such as in Lines 450-451, 460-466, 479, better say hydraulically functional or non-functional instead of living or dead tubes.

Cite this review as